# Established and Emerging Strategies for Drug Delivery Across the Blood-Brain Barrier in Brain Cancer

**DOI:** 10.3390/pharmaceutics11050245

**Published:** 2019-05-24

**Authors:** Alessandro Parodi, Magdalena Rudzińska, Andrei A. Deviatkin, Surinder M. Soond, Alexey V. Baldin, Andrey A. Zamyatnin

**Affiliations:** 1Institute of Molecular Medicine, Sechenov First Moscow State Medical University, 119991 Moscow, Russia; magdda.rudzinska@gmail.com (M.R.); andreideviatkin@gmail.com (A.A.D.); surinder.soond@yandex.ru (S.M.S.); alexeyvbaldin@gmail.com (A.V.B.); 2Belozersky Institute of Physico-Chemical Biology, Lomonosov Moscow State University, 119992 Moscow, Russia

**Keywords:** brain cancer, blood-brain barrier, drug delivery, FUS, CED, nanomedicine

## Abstract

Brain tumors are characterized by very high mortality and, despite the continuous research on new pharmacological interventions, little therapeutic progress has been made. One of the main obstacles to improve current treatments is represented by the impermeability of the blood vessels residing within nervous tissue as well as of the new vascular net generating from the tumor, commonly referred to as blood-brain barrier (BBB) and blood-brain tumor barrier (BBTB), respectively. In this review, we focused on established and emerging strategies to overcome the blood-brain barrier to increase drug delivery for brain cancer. To date, there are three broad strategies being investigated to cross the brain vascular wall and they are conceived to breach, bypass, and negotiate the access to the nervous tissue. In this paper, we summarized these approaches highlighting their working mechanism and their potential impact on the quality of life of the patients as well as their current status of development.

## 1. Introduction

Tumors of the central nervous system (CNS) account for about 3% [1,2] of the worldwide diagnosed neoplastic diseases and represent one of the most frequent causes of solid tumor-related deaths in childhood [3]. More than 85% of the CNS tumors affect the brain, which is also a primary metastatic site for tumors originating in other organs including the bladder, breast, kidney, and lung [4]. Gliomas are the most common tumors of the brain, and they can originate from different cell phenotypes that constitute the glia (astrocytes, oligodendrocytes, microglia, ependymal cells). Further categorizations are based on cancer aggressiveness which is evaluated on a scale ranging from grade I to IV, with grade IV being the most malignant, challenging to treat and likely to reoccur. In this scenario, treatments vary from simple observation for grade I glioma (with 5–15 years median survival) to surgical resection in combination with radio and chemotherapy for grade IV glioma (with 9–12 months median survival). Resection is by far the most effective treatment at least in terms of mass tumor reduction, but it is limited by the structural complexity and the primary function of the brain. Tumor debulking is usually referred to as “maximal safe resection” [5], implying a high risk of cognitive loss following the surgical procedure and incomplete removal of the tumor. Surgical limitations contribute to the high incidence of brain cancer recurrence, usually detected within 2 cm from the primary tumor [6].

Glioblastoma multiforme (GBM) is the most common tumor of the brain in adults, representing about 50% of all diagnosed primary brain cancers and usually classified as a grade IV glioma [7]. GBM is characterized by cellular and molecular heterogeneity that makes the optimization of the pharmacological interventions very difficult. The Stupp protocol is the gold-standard treatment for GBM [8], and it consists of surgical resection, postoperative radiotherapy, and temozolomide (TMZ), often used in association with adjuvant therapies including carmustine and PCV (procarbazine, lomustine, and vincristine). Despite their significant cytostatic properties in vitro, many Food and Drug Administration approved chemotherapeutics have shown limited curative benefits in the clinic. In the case of brain tumors, the development of more effective treatments is hampered by the specialized barrier function that characterizes the blood vessels residing in the central nervous system and usually referred to as the blood-brain barrier (BBB). In its physiological function, the BBB thoroughly selects and controls the mass transport occurring in and out the brain, limiting the healthy (and tumor) tissue diffusion of the administered pharmaceuticals while increasing the therapeutic doses in the patients that do not respond to the treatments is rarely a viable option. Also, the new blood vessels originating from the neoplastic lesions and often referred to as blood-brain tumor barrier (BBTB) are significantly less permeable than the neovasculature of the tumors developing in other organs being that their development is driven by the nervous system microenvironment. Herein, we describe new clinical and experimental approaches that aim to disrupt, bypass and negotiate these vascular barriers to favor the accumulation of therapeutics in brain cancer tissue.

### 1.1. Anatomy of the BBB: Tight Junctions

The very first researcher that introduced the concept of BBB was Lena Stern [9], a pioneer in the neuroscience field that coined the term *hematoencephalic* barrier to describe the BBB. Other scientists worthy of mention for their contribution to the discovery of the BBB’s functional and anatomical organization are Ehrlich, Lewandowsky, and Goldmann [10]. According to Sweeney et al. [11], the BBB is defined as “a continuous endothelial membrane within brain microvessels that has sealed cell-to-cell contacts and is sheathed by mural vascular cells and perivascular astrocyte end-feet.” In the human, the BBB characterizes over 100 billion capillaries that cover a total length of around 400 miles and a surface area of 20 M^2^ [12]. BBB vessels control the exchange of circulating molecules, nutrients and gas between the blood and the nervous tissue. In its physiological function, the BBB protects the brain from larger particles, proteins and hydrophilic molecules including potential neurotoxins and bacteria. It is believed that only 2% of small molecules and 0% of the large molecules can cross the BBB. Theoretically, only highly hydrophobic molecules with a molecular mass not higher than 400–500 Da can diffuse through this barrier [13]. BBB properties are due to many factors including (but not limited to) highly selective cellular sorting mechanisms regulating the transcellular traffic and the expression of tight junctions (TJs) between adjacent endothelial cells, limiting the paracellular transport.

TJs are composed of different transmembrane proteins including (but not limited to) the family of claudins, occludin, and junctional adhesion molecules (JAM-A, -B, and -C) and they interact with the cell cytoskeleton through membrane-associated guanylate kinases called zonula occludens proteins (ZO-1, ZO-2, and ZO-3). It is believed that all these proteins have a pivotal role in determining BBB function and a specific work performed on claudin-5 demonstrated that inhibiting its expression increased BBB permeability for molecules as large as 800 kDa [14]. This demonstration highlights the fine regulation that stands at the basis of BBB permeability, suggesting that TJ targeting could be a viable strategy to increase it. The efficiency of these proteins in closing the gaps between endothelial cells can be experimentally evaluated in vitro by measuring transendothelial electric resistance (TEER) that determines the resistance associated with ionic transport via the transcellular and the paracellular route. In the case of proper BBB reconstruction, TEER needs to be significantly higher (at least above 900 Ω×cm^2^) than in other endothelial settings (2–20 Ω×cm^2^). This value is considered the cut-off for the permeability of IgG, considering this under physiological conditions, TEER values range from 1500 to 8000 Ω×cm^2^ [15,16]. However, these values can vary as a function of the animal origin and the quality of the endothelial cells (primary or immortalized cell lines) [16]. Usually, immortalized cell lines do not provide TEER values higher than 200 Ω×cm^2^ while endothelial cells derived from inducible pluripotent stem cells can provide TEER values higher than 1500 Ω×cm^2^. Recent discoveries highlighted the possibility that, despite their sealing action, these proteins could determine two distinct mechanisms of BBB crossing. The first is known as “charge pore pathway’ in which the claudins form a molecular channel permeable only to small ions. The second is known as “size selective pathway” in which the passage to larger molecules occurs via a transient dissociation of TJ complexes [17]. A deeper understanding of these protein organizations could open new avenues of drug delivery as described later in the text.

### 1.2. Cellular and Enzymatic Elements of the Neurovascular Unit

The barrier function of the CNS endothelium is also determined by other cell phenotypes and biological structures including astrocytes, pericytes, microglia cells, neurons, and basement membranes which when taken with the endothelial cells, constitute what is commonly known as the neurovascular unit (Figure 1). Astrocytes are glial cells that interact with the endothelial cells through their polarized end-feet formations and control the BBB blood flow, development, and functions likely by enhancing the TJ expression in the mature BBB, even though they do not participate in its embryonic development [18,19]. In this context, some authors believe that astrocytes are not crucial for TJ expression, while others indicate that they can control TJ expression via Src-suppressed C-kinase substrates [20]. The modulation of BBB permeability occurs via secretion of important protein factors like the glial-derived neurotrophic factor, transforming growth factor-β1, basic fibroblast growth factor, interleukin 6, angiopoietin 1, retinoic acid, and Wnt [21,22]. Astrocytes also control the water exchange between intracellular, interstitial, vascular, and ventricular compartments by inducing the expression of the potassium channel kir4.1 and the water channel aquaporin-4. Pericytes have structural functions stabilizing the small BBB vessels and modulating the process of neovascularization and angiogenesis [23]. They are believed to significantly contribute to induce BBB gene expression as well as astrocyte end-feet polarization, even though more investigations are needed to reveal the complete spectrum of their activities in determining BBB and BBTB characteristics [24]. They control endothelial cell proliferation, survival, differentiation [18], and induce TJ mRNA expression in the embryonic formation of the BBB [25]. Microglia cells are the resident macrophages of the brain and contribute to the barrier function by modulating the innate immunity in the perivascular regions of the brain [22] and participating in the regulation of the expression of the TJ components [26]. Finally, neurons can induce the expression of TJ proteins like occludin and this phenomenon occurs synergistically with astrocytes [27]. BBB permeability also depends on enzymatic and immunological barriers limiting the molecular diffusion of blood solutes in the brain parenchyma. The endothelial cells composing the BBB express efflux transporters that are very efficient in transporting back to the luminal side the small hydrophobic molecules that crossed the BBB [28]. Efflux carriers are mostly adenosine triphosphate-binding cassette (ABC) transporters [27], and they are fundamental in clearing brain tissue from small lipophilic molecules. Between them, the P-glycoprotein (P-gp) and breast cancer resistance protein (ABCG2) were shown to have a significant role in the efflux of xenobiotics that penetrated the endothelial cell membrane, limiting the diffusion of chemotherapeutics in the brain parenchyma. P-gp is the most investigated pump, and its impact on brain transport was shown in knockout mice, where brain delivery increased up to 10–100 times [29]. This efflux pump is responsible for hampering the diffusion of many chemotherapeutics including doxorubicin (DOX), daunorubicin, vinblastine, vincristine, etoposide, and teniposide [30]. Also, together with the absence of endothelial fenestration, CNS endothelial cells showed a higher negative surface charge [31] and a lower transport rate through pinocytosis [32]. These parameters are highly considered for the designing and the development of more efficient delivery approaches (see later) since they constitute the physical and biological features of the BBB.

## 2. Models of BBB

One of the major obstacles in developing effective drug delivery across the BBB is the current lack of appropriate experimental in silico, in vitro and in vivo models allowing for cost-effective and high-throughput screening for different therapeutics. In silico models [33,34] of brain cancer are extensively developed for predicting tumor growth and infiltration in response to the treatments, while only a few cases are focused on predicting drug delivery in the brain neoplastic lesions [35,36]. The development of predictive computational models is critical in this field, also considering that mice have a brain structure extremely different from humans, counting for a 1:10 glial cell-to-neuron ratio versus a 1:5 ratio registered in humans [37]. Current in vitro and in vivo models are not reliable in mimicking and measuring BBB permeability respectively, but the research in this area is very active to discover new targets for favoring BBB accumulation as well as to understand the molecular dynamics that control TJ expression in the neurovascular unit. In the next sections, traditional and advanced methods to measure BBB function are described.

### 2.1. Traditional In Vitro Models of BBB

Three important parameters need to be consistent in establishing in vitro models of BBB: (1) low permeability validated through high TEER values, (2) expression of specific BBB biomarkers (i.e., TJ components and specific transporters and enzymes) [38], and (3) evaluation of barrier integrity through specific size molecular markers (sodium fluorescein, lucifer yellow, fluorescein isothiocyanate (FITC)-inulin, FITC-dextrans, and FITC- bovine serum albumin) [39]. In vitro models vary from simple acellular systems to very complex, multi-phenotype cellular models. Acellular models usually consist of parallel artificial membrane permeability assays (PAMPA) [40] and are based on synthetic lipophilic membranes that can only partially reproduce the physical properties of the BBB in vivo. These membranes are used to predict the passive diffusion of molecules through the barrier as a function of their hydrophobic or hydrophilic character. Few attempts to isolate brain capillaries and test BBB properties ex vivo have been performed, but the complexity of the isolation protocols, low reproducibility, and the difficulties to flow the tested molecules in the lumen of the isolated blood vessels affect their ordinary use [40,41]. On the other hand, new advances in cell isolation allowed for reconstructing the BBB with endothelial cells isolated from the brain, even though non-endothelial surrogate cellular models (i.e., Caco-2, ECV304) [42], that can still express TJs, are used for research purposes [40]. Many attempts at reconstructing the neurovascular unit were performed by co-culturing the endothelium with astrocytes, C6 glioma cells, pericytes, mixed glial cells, and conditioned media. Two-dimensional (2D) in vitro models are generated by seeding the endothelial cells on the apical side of a porous membrane while interacting with another cell phenotype (i.e., astrocyte or pericyte) seeded on the other side of the membrane via cellular protrusions extended through the pores. A third cell phenotype can be included in the system by seeding it on the bottom of the well to generate a conditioned culture environment and allowing for investigating the direct effect of cancer cells on endothelial cells forming the BBB [43]. The system can be further refined by coating the porous membrane with proteins belonging to the basal lamina and by decreasing serum concentration to favor the movement of the TJs from the cytoplasm to the basolateral region of the cells [44]. The serum can contain protein factors (i.e., vascular endothelial growth factor) that increase the permeability of the reconstructed endothelium in vitro, while supplementing the media with hydrocortisone or Adenosine 3′,5′-cyclic monophosphate (cAMP) analogs can increase endothelial barrier function since this second messenger is involved in maintaining the ultrastructure conformation of the TJs [44].

### 2.2. D Models and In Vivo Methods to Evaluate BBB Permeability

Three-dimensional (3D) models are currently one of the most advanced technologies to reconstitute in vitro the BBB, and are constituted of different cell phenotypes including cancer cells, normal astrocytes, and endothelial cells. The cells can assembly in spheroid units supported by hydrogels, scaffolds, and adhesion molecules. The group of Pasqualini developed 3D spheroids (1 mm in diameter) through magnetic levitation, by seeding glioma cells on a hydrogel composed by gold, magnetic iron oxide nanoparticles, and filamentous bacteriophage targeting cell integrins to favor cell interactions [45]. They showed that the spheroids could resemble in vitro the protein expression of tumor biomarkers (*N*-cadherin) registered in vivo and that multiple cell phenotypes could be mixed in the same spheroid unit to investigate cell interaction, biology, and drug diffusion while providing effective implantable tumors. As it occurs in vivo, a necrotic core characterized the spheroids and, by modulating the external magnetic field, it was possible to control their size and shape. Also known as organ-on-chip, new advances in microfluidic devices were utilisied to better recapitulate the characteristics of the BBB tissue by combining geometrical, physical, and biological features of this tissue [46,47]. These tools can also be implemented with sensors providing real-time and continuous measurements of the changes occuring in BBB permeability under different conditions. These systems usually consist of polydimethylsiloxane that provides optimal integration with microscopy analysis and fine-tuned engineering via soft lithography on the microscale, which supports the organized culturing of cellular layers derived from the nervous tissue (i.e., endothelial cells, neurons, and astrocytes). In addition, they can be integrated with channels in which the media flows and supports the growth of endothelial cells to mimick the characteristics of primary tissue [48,49]. The different compartments allow for intercellular interactions to establish the critical cues of cellular communications for generating a functional BBB in vitro. In this scenario, the generation of refined 3D models can represent a breakthrough in the development of more advanced tools to investigate the biology of the neurovascular unit since they can: (1) include multiple interacting cell phenotypes and (2) evaluate BBB in flow conditions. However, to date, these systems are too complex to be ordinarily used worldwide and drug screening is still mostly performed in traditional transwell systems. For more information about these systems, we suggest the following reviews [16,50].

In vivo pharmacokinetic evaluation in the brain depends on different biological parameters including blood flow in the BBB, the density of influx and efflux transporters as well as the affinity of the drug for these transporters. The goal of these measurements is to quantify the product between the amount of therapeutic that crossed the BBB and the surface area of the BBB [51]. In vitro pharmacokinetics methods are not considered reliable because drug passive diffusion is generally over-estimated, while the active transport is frequently underestimated [52]. Different advanced techniques allow for calculating drug accumulation in the brain parenchyma like ex-vivo equilibrium dialysis performed on brain homogenates or slices or by using dialysis fibers directly implanted in vivo. This second method is generally preferred when possible because it allows for measuring drug concentration in the brain in the presence of normal blood flow. Also known as brain microdialysis, this method consists of implanting a small capillary in the brain parenchyma under continuous perfusion (Figure 2). The tip of the capillary is semipermeable and allows for collecting tissue fluids. However, the insertion of the capillary in the brain parenchyma could damage the BBB continuity with consequent leakage of blood fluid leading to an overestimation of the drug concentration. Overall there are three significant challenges in increasing brain drug delivery: (1) targeting the vasculature of the brain, (2) overcoming the BBB, and (3) favoring drug diffusion in the brain diseased tissue. In the next chapters, available information about current strategies for crossing the BBB will be described with a focus on their working mechanisms as well as the pros and cons of the different methods.

## 3. Breaching the BBB

Considering the importance of the brain and the physiological relevance of the BBB, barrier disruption by affecting TJ integrity and/or endothelial cell continuity has to be fine-tuned and reversible. These properties are fundamental because potential extravasation of circulating factors (i.e., albumin) can be very toxic for the neurons [53]. Traditional approaches to transiently affect BBB integrity are based on the injection of a hyperosmotic solution (usually consisting of a highly concentrated solution of mannitol [54]) just before the administration of the therapeutics. Hyperosmotic solutions can induce endothelial cell shrinking with a consequent increase in vascular leakage in the brain parenchyma. This approach was effective in increasing the overall survival of the patients (from 11 to 17 months), but it requires repeated hospitalization and is also considered very invasive (it needs patient sedation), unspecific, and accompanied by severe systemic toxicity, including neurological deficits, strokes, seizures, and new tumor-nodule formation [55]. Current clinical trials are devoted to optimizing the use of hyperosmotic solution based on mannitol [56] or NaCl [56] to increase chemotherapy and antibody delivery to the brain tumor and decrease intracranial pressure. Recently it was shown in rats that the osmotic disruption of the BBB (achieved via intracarotid injection of a 25% solution of mannitol) could be exploited to increase the delivery of hydrophobic siRNA, previously modified with phosphocholine (PC)-docosahexanoic acid. The increase in the hydrophobicity of this biological therapeutic was shown to enhance the retention of the siRNA in the brain without affecting its therapeutic action. The group of Chung developed a polymeric carrier of polydixylitol with high osmotic power that showed high efficiency in nucleic acid delivery in vitro and in vivo. More importantly, they showed that the osmotic BBB opening could induce caveolae-mediated transcytosis of the carriers while having a low toxicity profile [57]. More advanced methods to breach the BBB are described in the following sections.

### 3.1. Focused Ultrasounds

Advanced options available to breach the BBB consist of physical mechanisms that can be remotely applied with low invasiveness. Ultrasounds were shown to be effective in increasing the BBB permeability and implants able to deliver fine-tuned acoustic pressures in the brain are currently under clinical trial evaluation [58]. Ultrasound therapeutic potentialities were known since the 1940s, but it was only recently that technical ameliorations to this technique in avoiding skull overheating and in improving energy transfer have made it extremely non-invasive and transformational [59]. The control of the applied forces is critical considering that current brain cancer treatments (i.e., TMZ) require multiple drug administrations. In this effort, the transfer of the acoustic energy can be compared to the use of a magnifying glass that converges high levels of light to ignite one small area (focal point), while outside this point the ultrasound can penetrate the tissue with no effect (Figure 3). Focused ultrasound (FUS) showed the most promising results when used in combination with microbubbles representing one of the most advanced ways to breach the BBB safely. As it is reversible, the effect lasts for a few hours [60,61] and the treatment can be targeted to some brain regions sparing healthy nervous tissue sites [59]. Microbubbles (5–10 micrometer in diameter) usually consist of lipids, polymers, surfactants, or proteins like albumin, and are loaded with gases like perfluorocarbon. These materials usually have amphipathic characteristics, and their stability depends on the formation of hydrophobic, covalent, and disulfide bonds, respectively [62]. Microbubbles are currently commercially available for diagnostic purposes in echocardiography (Optison^®^ Definity^®^, and Sonovue^®^ microbubbles) since when exposed to ultrasounds their vibration generates a strong echogenic signal based on the difference in acoustic impedance between the gas and the surrounding tissue. Today there are at least three clinical trials focused at testing FUS to improve the conditions of patients affected by GBM. To evaluate the safety of this approach, the group of Liu exposed rats to repetitive FUS at three different acoustic pressures defined as a function of erythrocyte extravasation and at different microbubble doses. The study concluded that high acoustic pressure and microbubble doses could cause brain hemorrhage, tissue necrosis, cell apoptosis, astroglial activation, and glial scarring [63]. However, moderate acoustic pressures and adequate microbubble administrations allowed for a safe breach of the BBB. This work is fundamental in the field because it demonstrates the need to finely tune the conditions of the system to avoid tissue damage [63]. In support of this data, repetitive FUS applications were applied on primates by using an implantable ultrasound device to evaluate the long term effects of this procedure. The system was used in combination with microbubbles, and the animals were exposed to ultrasounds for a total of seven times in four months. Magnetic Resonance Imaging (MRI) was used to assess the successful breaching of the BBB and positron emission tomography coupled with fluorine-18-labeled fluorodeoxyglucose detection did not show any changes in glucose metabolism. At the end of the experiment, the animals did not show any neurological distress and histology showed limited extravasation of red blood cells [64]. Different hypotheses on the mechanism at the base of BBB breaching were elaborated. High acoustic pressures induce inertial cavitation with the generation of heat and microbubble collapse allowing for a controlled microdamage of the brain vasculature (sonoporation) through the formation of heat, shock waves, and microjets. Experimental data are showing that BBB opening can occur at the level of the TJs and that this procedure likely increases paracellular permeability following endothelial spasm. On the other hand, relatively low acoustic pressures can generate stable cavitation of the microbubbles where the carriers oscillate in a nonlinear fashion without destruction. This phenomenon occurring in the proximity of the endothelial bed increases the flow dynamics of the liquid that surrounds the microparticles increasing shear stress forces (micro-streaming) [65] that can modulate the BBB ion channel and receptor activity [66] and induce caveolae-dependent transport [59,67] in brain parenchyma. FUS showed to increase the delivery of a plethora of anticancer therapeutics, including chemotherapeutics, antibodies, small interfering siRNA, and nanoparticles like superparamagnetic iron and gold nanocarriers as theranostic tools [7]. Ultrasound applications can also favor nanoparticle diffusion in the brain parenchyma by increasing the physiological porosity of the extracellular space, where the tissue architecture depends on electrostatic bonds connecting cells and extracellular matrix. The pioneering work of the group of Frenkel [68] demonstrated on ex-vivo brain slices that pulsed ultrasounds can create pores up to 500 nm in the perivascular space, potentially favoring therapeutics and nanoparticle diffusion after BBB crossing, even though in vivo experimentations are necessary to confirm this data. Also, it was shown that this approach could locally decrease the BBB expression of P-gp following a mechanism probably related to changes in blood flow [30]. Current limitations of FUS are represented by the short half-life of the microbubbles (average circulation time is estimated in 3–15 min) that are readily entrapped in the organs of the mononuclear phagocytic system due to their size in the microscale [69]. Other limitations are related to the need for coupling FUS with constant imaging monitoring to avoid major side effects and better target the tumor tissue.

### 3.2. Photodynamic Therapy

The photodynamic effect is another example of locally increased BBB permeability. This approach is based on light irradiation of photosensitive molecules (i.e., 5-aminolevulinic acid) and it was first developed for ameliorating current imaging and surgical techniques through intraoperative irradiation. Also in this case, the treatment can be focused to a minimal area of the brain and, compared to the healthy tissue, it was shown that some photosensitizers have a natural tropism for the neoplastic lesions [70]. Photodynamic irradiation can increase the delivery of large molecules and nanoparticles [33] while its invasiveness can be reduced through the generation of efficient optical clearing windows in the skull tissue to avoid the high light scattering generated by the bone. Tissue optical clearing windows are obtained by immersing the naked skull in hyperosmotic solutions before irradiation [71] or by implanting irradiation devices. A recent work of the group of Zhu demonstrated that this approach increases reactive oxygen species generation that in turn induce tight (CLDN-5 and ZO-1) and adherent (VE-cadherin) junction protein internalization with a consequent decrease of the barrier function. This evidence is fundamental because they confirm the reversibility of the BBB breaching through the rearrangement of the junctions while providing a working mechanism of this procedure.

## 4. Bypassing the BBB

There are essentially two extensively investigated pharmacological approaches that can be referred as to interstitial treatments for brain cancer: the application of biodegradable wafers and convection-enhanced delivery (CED), and both are designed to bypass the BBB. Generally, they are considered extremely invasive; however, both are already included in the clinical practice even though a lot of research is still dedicated to increasing their therapeutic benefits and their safety.

### 4.1. Biodegradable Wafers

Biodegradable wafers were designed to exploit the surgery step in which the tumor is removed. As aforementioned in the text, in the case of brain cancer, it is impossible to perform massive tissue debulking and the cavity that results from cancer tissue removal is usually still positive for the presence of cancer cells. In this scenario, therapeutic wafers are implanted in the area of the tumor bed, where they locally release chemotherapeutics killing the residual cancer tissue. Gliadel^®^ wafers represent the gold standard therapeutic of this approach. Gliadel^®^ consists of the copolymer polifeposan (1,3-bis(p-carboxyphenoxy) propane and sebacic acid, 4:1 molar ratio) [72] loaded with the alkylating agent carmustine (3.85% *w*/*w*). The wafers are disk-shaped of 14 mm in diameter and 1 mm in thickness, with an overall weight of 200 mg. A previous investigation in primates (without tumor or tissue removal) receiving the wafers demonstrated that carmustine can diffuse in the brain parenchyma between 2 and 8 mm from the implantation site [73]. These wafers are FDA approved in many countries and they are designed for being completely biodegradable and support a sustained release of at least five days after implantation [74], while the wafer is supposed to be completely degraded in 2–3 weeks. The disks are directly applied on the surface of the brain cavity where the tumor was removed and up to eight disks can be implanted (Figure 4). This number can sometimes offer limitations in the efficacy of Gliadel^®^ wafers, because eight disks may not cover all the area of the tumor cavity. For this reason, the surgery needs the support of intraoperative frozen section diagnosis [75] to apply the disks in the areas of the cavity where more neoplastic tissue is detected. Different studies in the last decades demonstrated that, compared to the patients that received only placebo treatment, Gliadel^®^ was efficient in increasing, by several weeks, the overall survival of the patients. In a typical clinical scenario, Gliadel^®^ application is followed by the Stupp protocol and it is not indicated for patients with not resectable tumors or with tumors infiltrating the ventricular system, which still represent the majority of the cases. Major concerns about this treatment are related to the potential adverse effects that the wafers can generate (i.e., cognitive loss, seizures, poor wound healing, intracranial hypertension, cerebrospinal fluid leakage, and cyst formation) via the formation of carboxylic acids as a byproduct of the polyanhydride polymer. Additionally, the process of drug loading involves the presence of toxic residues (several hundred parts per million) like dichloromethane, acetone, tetrahydrofuran, and ethyl acetate. Edema, in particular, frequently occurs (from 6.8% to 25% of the cases [76]) in patients that received Gliadel^®^ and it was speculated that the toxicity could derive from cell necrosis of the neoplastic tissue sensitive to carmustine. Usually, the edema is refractory to corticosteroids treatments, while a controlled use of bevacizumab (anti-VEGF-A treatment) just after the surgery was shown to reduce the risk of these side effects [77]. Infection is another documented potential adverse effect of the wafers even though a large study performed by the group of Chaichana, demonstrated that the previous presence of diabetes mellitus, multiple resections, and prolonged hospitalization are probably the main causes at the base of this phenomenon [78]. In addition, the group of Carpentier described a clinical case in which they analyzed not fully degraded wafers removed from the brain of a patient that underwent a second surgery due to a tumor recurrence [73]. The analysis demonstrated that incomplete degradation occurred at the level of the sebacic acid units (hydrophilic material), causing the scientists to speculate that a hydrophobic layer of unspecified biological material generated on the surface of the wafers reducing their degradation. In an attempt to define the patient population that more likely could benefit from Gliadel^®^, the group of Urbschat demonstrated that high expression of miRNA-181d is associated with low overall and progression-free survival of the patients [79], even though more investigations are necessary to understand the mechanism of this phenomenon. Current research focuses on generating implantable systems that can overcome Gliadel^®^ limitations like extending this approach to more therapeutics and prolonging the drug release period. In this scenario, materials like biocompatible silk [80] and poly(lactic-co-glycolic acid) (PLGA) [81] represent optimal candidates to advance this kind of technology since they can be loaded with different therapeutics and their ultrastructure can be tuned to achieve the fine controlled release of different drugs [82].

### 4.2. CED

CED allows for intra-tumoral local drug infusion via a catheter placed directly into the tumor parenchyma. Despite its invasiveness, CED was evaluated in clinical trials for the treatment of GBM and it demonstrated an adequate safety profile for several convection-delivered agents [83]. However, in vitro and in vivo experimentations unveiled that the convective flow can favor glioma and cancer stem cells invasion via the activation of the CXCR4-CXCL12 signaling pathway. The binding of chemokine CXCL12 to its receptor triggers multiple responses including increasing in intracellular calcium flux, gene transcription, chemotaxis, cell survival, and proliferation [84]. To prevent this side effect, it was recently shown that the co-administration of the CXCR4 antagonist AMD3100 can decrease this phenomenon [85]. Another limitation related to CED is the occurrence of backflow during the treatment, described as the fluid discharge around the catheter shaft with consequent leakage of the therapy out of the brain instead of into the nervous tissue. Backflow-free catheters are under evaluation to increase the flow rates at higher values to mitigate this phenomenon [84]. CED was investigated for the delivery of various agents, such as conventional chemotherapy [86], cytotoxin-ligand conjugates targeting cell surface receptors [87], antisense oligonucleotides [86,88], and nanovectors [86,89]. CED showed promising results when used to infuse brain-penetrating nanoparticles (BPNP) that resulted in significant tissue penetration thanks to their small size and their stability as monodispersed agents and allowed for drug release in a controlled fashion. BPNP are PLGA nanoparticles loaded with paclitaxel and modified with nile red to permit their imaging. The presence of polyethylene glycol (PEG) was shown to favor their diffusion into the tumor parenchyma highlighting the need to improve the design of the therapeutics that are locally injected to support their distribution in tumor tissue [90].

## 5. Negotiation of the BBB

New approaches of drug delivery aimed at negotiating the passage through the BBB have been proposed based on current knowledge of the transport mechanisms used by this specialized endothelium. Some of them exploit the physical properties of the BBB; others are based on the BBB biochemical receptor and transporter profile. To date, three main routes of BBB negotiation have been developed and referred to as adsorptive-mediated transcytosis (AMT), transporter-mediated transcytosis (TMT), and receptor-mediated transcytosis (RMT) (Figure 5). In this effort, the development of rationally designed nanocarrier surface modifications was shown to be useful to exploit these transport routes. Also, nanomedicine provided a mean to protect the encapsulated drug in the blood environment as well as to increase its bioavailability.

### 5.1. AMT

The highly negative surface charge of the BBB surface can be exploited to deliver in the brain parenchyma molecules like cell-penetrating peptides (CPP) that are composed of positively charged and amphipathic amino acids. The transactivator of transcription (TAT) [91] derived from the human immunodeficiency virus and the peptide gH625 [92] derived from the glycoprotein H of herpes simplex virus type are typical examples of CPP used for brain delivery. CPP can be directly conjugated with a therapeutic or applied on the surface of nanocarriers to favor their transport across the BBB. Peptide design is often associated with activable structures that can be sensitive to the acidic pH or to the action of metalloproteases, typical features of tumor environment providing additional cues for targeting [92]. The surface of nanoparticles can also be chemically modified to be positive like in the case of albumin nanoparticles loaded with DOX [93]. Given these properties, AMT allowed for the successful delivery of many biologicals (including nucleic acids) and chemical therapeutics. In the pioneering work of the group of Sabel, three parameters were related to nanoparticle AMT, including size, surface charge, and hydrophobicity (by incorporating surfactants in the structure of the particles) were evaluated [94]. In this study, they used polybutylcyanoacrylate nanoparticles as a carrier model, evaluating their diffusion across an in vivo model of a blood-retina barrier. Contrary to other findings, demonstrating that size (indirectly proportional to nanoparticle ability to overcome BBB) and surface charge (positive particles can better cross BBB) are fundamental in designing nanocarriers for BBB, they showed that hydrophobicity was the key to overcome this endothelium. In particular, they demonstrated that non-ionic surfactants have a higher impact on BBB permeability than anionic ones since they favor the occurrence of an apolipoprotein E corona that was previously shown to increase BBB nanoparticles incorporation via RMT [95,96]. This work is important for the future development of drug delivery systems for the brain because it demonstrated that the surface modifications that occur in the blood milieu are the real key players in determining the brain accumulation of the carriers. However, targeting strategies based on AMT are considered very unspecific since they aim to target some physical features of the BBB that are present throughout all the vascular system, and their internalization can efficiently occur also in off-site organs with a consequent decrease in treatment efficacy and potential occurrence of side effects.

### 5.2. RMT and TMT

These approaches exploit membrane receptors that are generally over-expressed on the surface of the BBB. However, as highlighted by Warren [97], the BBB is not a static structure, as it can differentially modulate its permeability in physiologic and pathologic conditions. Experimental and clinical data are showing that in different areas of the brain, the BBB can be extremely variable in terms of permeability and expression of transporters, influx, and efflux pumps [98]. TMT and RMT have transformative potential in the generation of new drug delivery strategies for brain cancer because, despite their role in the molecular transport of the brain, they are generally overexpressed on the BBB, representing an optimal target for this tissue. Even though receptors and transporters have very different biological functions, their relevance in drug delivery is similar since they negotiate the internalization of the therapeutic (or the carrier) via interaction with specific ligands and for this reason they are dealt in the same section of this review. TMT is usually referred as the transport route allowing for the passage across BBB of small polar nutrients like sugars, vitamins, hormones, and amino acids [99]. In this effort, nanoparticles can be functionalized with mannose to overcome the BBB via GLUT1 [100] or with quaternary ammonium to exploit the choline transporter [101]. The group of Lu recently developed a method to generate a thin layer of polymeric acetylcholine and choline analog as a coating around individual proteins to favor their delivery in the brain parenchyma by exploiting choline transporters [102]. The coating is generated via a biodegradable crosslinker to facilitate the release of the proteins in the brain parenchyma. With this method, they showed the successful delivery of different proteins including the antineoplastic agent rituximab. On the other hand, RMT is referred to drug delivery strategies exploiting receptors that favor the transport across the BBB of larger proteins. To this category belong the transferrin receptor (TfR) [103,104], low-density lipoprotein receptor (LDLR) [105], diphtheria toxin receptor [106], and nicotinic acetylcholine receptor [107]. TfR has been extensively investigated in the field because it is overexpressed in glioma cells and the BBB, while it is not expressed in the blood vessels of other tissues [103]. In this context, TfR targeting can be functional both at the level of the BBB and at the level of the cancer cells which have undergone further extravasation to the brain parenchyma. The antibody for TfR OX26 was shown to favor BBB transcytosis of therapeutics [108] and nanoparticles [109], but recent evidence contradicted this data, highlighting the need of more research to understand this phenomenon. TfR targeting was recently used also to deliver theranostic agents [110] and to modify liposomal nanoparticles in combination with p-aminophenyl-α-d-manno-pyranoside, targeting GLUT-1 to achieve a double targeting of BBB and cancer cells [111]. RMT also represents a viable strategy to deliver antibodies across the BBB. The antibodies can be conjugated with therapeutics exploiting the RMT trafficking to favor drug delivery across the BBB [112]. To this end, many investigations were performed to understand the optimal antibody affinity towards the targeted receptor and in the case of TfR it was shown that a weaker binding improved antibody delivery by avoiding receptor dimerization and internalization to the endolysosomal compartment [113,114]. In this scenario, recent advances in the field have allowed for the generation of bispecific antibodies capable of recognizing two different targets, usually represented by a targeting receptor and a tumor molecular target [115]. Bispecific antibodies are under consideration also for redirecting T cell specificity towards cancer lesions, including malignant glioma [116]. Compared to TMT, RMT manages the transport of larger molecules, but it is fundamental to state that neither of these transport mechanisms have evolved to negotiate the passage of nanocarriers and there is no evidence to support that these receptors can physically mediate nanoparticle transcytosis. However, it was shown several times that their targeting can also increase nanoparticle trafficking to the abluminal side, therefore more investigation in this area is necessary to dissect the working mechanism of this kind of transport.

### 5.3. Opportunities in Targeting Endothelial Junctions

Targeting the proteins involved in the formation of adherens junctions can be useful to increase the transport across the BBB. These approaches are usually described as methods of BBB disruption, but we believe that they better fit in the BBB negotiation section because (1) the opening effect is relatively shorter and tunable compared to other methods of BBB disruption [117] and (2) they rely on specific biochemical interactions targeting junction stability like HAV6 and ADTC5 peptides designed to interact with cadherins [118,119]. The group of Siahaan demonstrated that it is necessary to take into consideration peptide stability in the biological serum as well as their structural rigidity to enhance the junction-peptide interaction. In particular, they showed that the use of cyclic versions of a peptide could provide better results than the linear amino acid sequences [117]. In the last two decades, peptidomics studies allowed for generating peptides interacting with claudin-1 (i.e., C1C2 [120]) or occludin [121] (i.e., OCC1 and OCC2). Interestingly it was shown that the destabilization of these proteins could favor their internalization and cellular recycling in combination with a parallel decrease of their mRNA expression [120]. Claudin and cadherin regulation can also be achieved via RhoA signaling activated by the A2A adenosine receptor [122]. Angubindin-1 is a peptide (200 amino acids) derived from the iota toxin of *Clostridium perfringens* and able to bind angulin-1 and -3, known to destabilize the proteins of the tight junctions [123]. After intravenous injection, it was shown to increase BBB permeability probably by increasing the size-selective pathway, and enhancing the delivery of antisense oligonucleotides with no toxic effects [17,123]. Recent evidence showed that the family of the lysophosphatidic acid receptors (LPARs) is overexpressed in the CNS microvasculature and more specifically LPAR1 is overexpressed in the brain vasculature [124]. Upon interaction with lysophosphatidic acid, this receptor can increase the permeability of the BBB possibly via RhoA activation. Intravenous treatments with lysophosphatidic acid can transiently (20 min post administration) increase BBB permeability for small and large molecules including Gd-DTPA, the infrared dye 800cwPEG, and Rhodamine 800 (often used to measure the activity of the efflux pump P-gp) [125,126]. More importantly lysophosphatidic acid was efficient in triplicate negative charged iron oxide nanoparticle (estimated size 30 nm) brain deposition favoring their potential use as theranostics [127]. Histamine [128] and bradykinin (BK) [129] and its more stable compounds [130] can modulate TJ expression via the beta receptor expressed on the endothelial cells while controlling the intracellular concentration of Ca^2+^, in particular in the cells composing BBTB [131]. This ion is at the base of the TJ ultrastructure and when released in the cell cytoplasm can affect their tightness probably by increasing the expression of e-NOS and n-NOS with the parallel decrease in nuclear expression of the transcription factor ZONAB and decrease of the mRNA and protein expression of claudin-5 and occludin [131] (Figure 6). On the other hand, the zonula occludens toxin (ZOT) [132] was shown to affect TJ stability through a mechanism dependent on phospholipase C and protein kinase C. These enzymes can directly affect actin cytoskeleton reorganization and in turn, increase BBB permeability to chemotherapeutics like paclitaxel.

## 6. Crossing Blood-Brain Tumor Barrier

Unlike the BBB, the BBTB has to be considered a pathological tissue since it is the product of the neoplastic lesion. Compared to regular BBB, BBTB is generally considered more permeable, even though as aforementioned, its barrier function (estimated cut-off of around 12 nm) [133] is significantly higher than what usually registered for the neo-vasculature generated from tumors in other organs. Even though the leaky behavior of BBTB can be appreciated also through regular MRI via brain edema detection, its dysfunction is not homogenous in the tumor tissue [28], and high functional variability was also appreciated between different patients. In the case of BBTB, the investigation of peculiar surface markers overexpressed in this tissue represent the best strategy to design carrier targeting, because it provides the opportunity to target the pathological tissue specifically. Despite the traditional targets described for BBB, BBTB can theoretically be targeted exploiting the typical surface biomarkers of growing blood vessels. For example, it was shown that targeting integrin ανβ3 through the cyclic RGD peptide applied on the surface of polymeric polylactic acid and polyethylenimine particles [134] increased the brain delivery of encapsulated nucleic acids and paclitaxel, respectively, when compared to non-functionalized carriers. Recent findings also demonstrated that brain drug delivery could benefit from strategies aimed at normalizing pathological vasculature like administration of the Ang2-binding and Tie2-activating antibody [135]. More importantly, the group of Koh demonstrated that this approach could eventually enhance brain drug delivery by decreasing the interstitial pressure while increasing blood vessels perfusion and tissue oxygen levels modulating immune cell infiltration [135]. The group of Moses analyzed commercial GBM cell lines and 70 tumor samples from patients affected by GBM and identified the integrin α2 (ITAG2) as a novel surface biomarker for this BBTB. This integrin is involved in cell migration and the surface functionalization of DOX-loaded liposomes (with an antibody specific for this protein) showed cytostatic effects in vitro and in vivo, highlighting the importance of more research in the discovery of novel endothelial surface biomarkers for the treatment of brain tumors [136]. Compared to the surrounding healthy tissue, a brain tumor is characterized by significant changes in cell metabolism tissue that can represent an important targeting cue. Albumin, for example, is normally excluded from the brain parenchyma by the presence of the BBB, but it was shown that neoplastic lesions can increase its uptake likely to exploit this circulating protein as a source of amino acids. The group of Huang demonstrated that brain cancer overexpressed secreted protein acidic and rich in cysteine (SPARC) and GP-60, increasing the albumin endothelial transcytosis and cancer uptake, respectively (Figure 7). To target these receptors, they generated albumin nanoparticles (100 nm) encapsulated with paclitaxel and fenretinide and modified their surface with a CPP to favor particle diffusion in the brain parenchyma [3]. Finally, it is worth mentioning that brain tumors can generate new blood vessels via vascular mimicry, a phenomenon that can occur as a drug resistance mechanism upon the use of anti-angiogenic adjuvant therapies [137]. Both in human and in pre-clinical models, it was shown that the presence of red blood cells within vessel walls lined up with cancer cells and basal lamina. These cells were positive to periodic acid-Schiff but negative to CD34 immune staining, excluding their endothelial nature. In this case, further investigation is necessary to understand the advantages of targeting vascular mimicry and potential therapeutic effects of this approach.

## 7. Cell and Gene Therapy

Approaches based on local delivery were used to inject healthy neural stem cells [138,139] to exploit their ability to infiltrate neoplastic lesions in the CNS. Genetically modified neural stem cells can be manipulated to generate and release cytotoxic molecules including prodrug-activating enzymes, apoptosis-inducing agents, antibodies [140], and oncolytic viruses [141]. The group of Portnow used neural stem cells modified for expressing cytosine deaminase to convert the prodrug 5-fluorocytosine (that can cross the BBB) to 5-fluorouracil. They directly injected the cells close to an established glioma or in the opposite hemisphere and they showed successful infiltration of the stem cells in the tumor parenchyma as well as higher cytostatic properties upon treatment with the prodrug [142]. Unfortunately, this procedure is affected by low efficiency in implanting viable cells. A way to avoid this issue is to seed the cells in vitro on a biocompatible scaffold (i.e., fibrin) and, like in the case of Gliadel, to insert the scaffold in the cavity obtained after brain tumor removal [143]. In this scenario, HEK 293 EBNA modified to release endostatin were encapsulated in an alginate scaffold prior to brain implantation, inhibiting in vivo GBM-induced angiogenesis process [144], while polymeric biodegradable scaffolds seeded with stem cells overexpressing secretable tumor necrosis factor apoptosis-inducing ligand were implanted to inhibit brain tumor growth [145]. Recent advances in biological drug delivery systems demonstrated that neutrophils could be exploited to overcome the BBB and increase drug delivery for brain cancer. The group Zhang loaded neutrophils in vitro with cationic liposomes and, after systemic administration, they infiltrated the neoplastic lesion guided by inflammatory cytokines and chemokines. The authors loaded the carriers with paclitaxel (which compared to other chemotherapeutics showed a minor impact on neutrophils biology) and exploited the cytokine gradient induced by the surgical removal of the tumor, exactly reproducing the clinical scenario [146]. Other biological agents used to treat brain cancer are adeno-associated viruses (AAV) since they are safe, effective, and one of the most promising methods to enhance gene delivery through the BBB [147,148]. The ability of different serotypes to effectively overcome the BBB is well known [149] even though the mechanism used to overcome BBB has still to be elucidated [150]. Engineering efforts have yielded several AAV variants that can efficiently transduce the CNS via systemic delivery in adult mice [151]. The group of Gao [149] tested nine different AAV vectors encoding green fluorescent protein (GFP) (injected into the superficial temporal vein of the mice) showing that they could increase GFP intensity in different brain compartments. Recently AAV targeted evolution technique revealed a novel recombinant AAV-PHP.B that transfers genes throughout the CNS with an efficiency that is at least 40-fold greater than that of the natural viruses [152]. Despite the optimistic perspectives regarding AAV-based delivery, some drawbacks need to be considered. The insert capacity of the vector is limited by 4 kb due to the AAV nature limiting some possible implementations of this method. In addition, the immune response to viruses can dramatically decrease the efficiency of gene transfer by systemic delivery. On the other hand, virus-induced expression of transgenes in the central nervous system can last for years [153], while the ectopic expression of the transgene can cause side effects. This limitation can be overridden by using cell type-specific promoters [151].

## 8. Summary

Brain cancer is characterized by an extremely fast and lethal outcome. While radiotherapy and surgery represent viable options to treat this condition, current pharmacological interventions are inadequate to increase patient survival as well as to improve their quality of life. Preclinical experience suggests that current chemotherapeutics are very effective on brain cancer growth, but unfortunately their delivery is affected by the BBB and the BTBB. Finding new pharmaceutical strategies to increase drug delivery in the brain is of primary importance considering that this condition is more frequent in the elderly and that by 2050, more than 1.7 billion people will be over 60 years old [2,154]. Current trends in drug delivery aim to bypass, breach, and negotiate the BBB. Between them, the use of hyperosmotic solutions [56], FUS [155], biodegradable implants [156], and CED [157] are extensively used and/or investigated in the clinic for their safety and efficacy. On the other hand, to our knowledge, only one model of targeted nanoparticle is under investigation in humans for the treatment of GBM [158]. While these strategies were shown to be effective, currently they can increase overall patient survival only by a few weeks. Future perspective needs to take in consideration the complexity of the BBB by developing improved models to study the biology of this tissue as well as the variability within brain cancer tissues [97], including the regional and country variation registered in the incidence of this disease [159]. These data induce to believe that the development of more effective therapeutic strategies should take into consideration the combination of different delivery approaches, possibly with the support of improved imaging techniques, whose role is fundamental to target the cancer tissue.

## Figures and Tables

**Figure 1 pharmaceutics-11-00245-f001:**
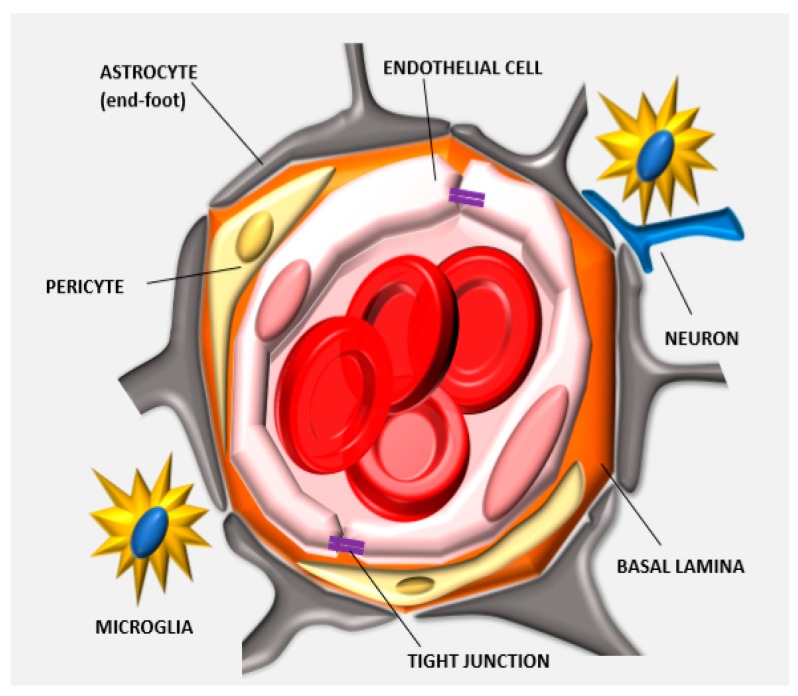
Anatomy of the neurovascular unit: the blood-brain barrier (BBB) structure is determined by different biological components that organize together in forming the neurovascular unit. Endothelial cells form the lumen of the capillary, interact with the basal lamina and the pericytes embedded in this matrix. The astrocytes, neurons, and microglia cells further support this cellular backbone. Other physical agents determining the barrier function of this specialized endothelium are the tight junctions (TJs) that are expressed between adjacent endothelial cells.

**Figure 2 pharmaceutics-11-00245-f002:**
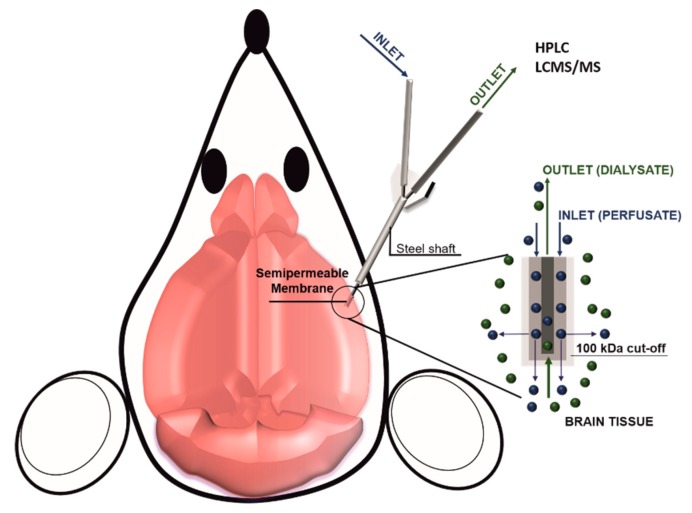
Scheme of brain microdialysis: A catheter is inserted in the brain tissue, while a controlled system (i.e., a syringe pump) injects in the brain a perfusate solution. At the end of the catheter is applied a semi-permeable membrane that allows for the injection of the perfusate, as well as for the collection of the dialysate composed by the perfusate and the brain tissue fluids. The collected dialysate can be eventually analyzed for its molecular content.

**Figure 3 pharmaceutics-11-00245-f003:**
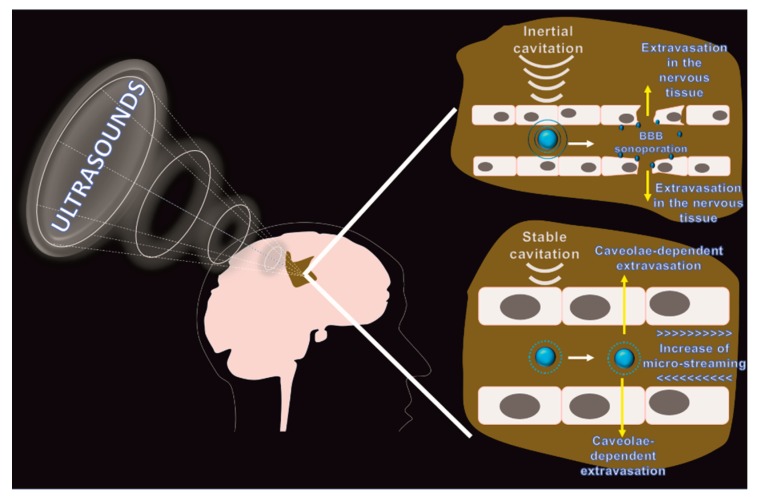
Effects of focused ultrasound on BBB permeability: External FUS is applied in combination with microbubbles injection. As a function of the acoustic pressure applied it can be obtained by: inertial cavitation and sonoporation, where relatively high levels of acoustic pressures induce the microparticles to collapse with a consequent controlled breaching of the BBB; and stable cavitation and micro-streaming where relatively low levels of acoustic pressure induce the microbubbles to vibrate increasing the flow shear stress in the proximity of the vascular wall with a consequent increase of the endothelial transport through caveolae.

**Figure 4 pharmaceutics-11-00245-f004:**
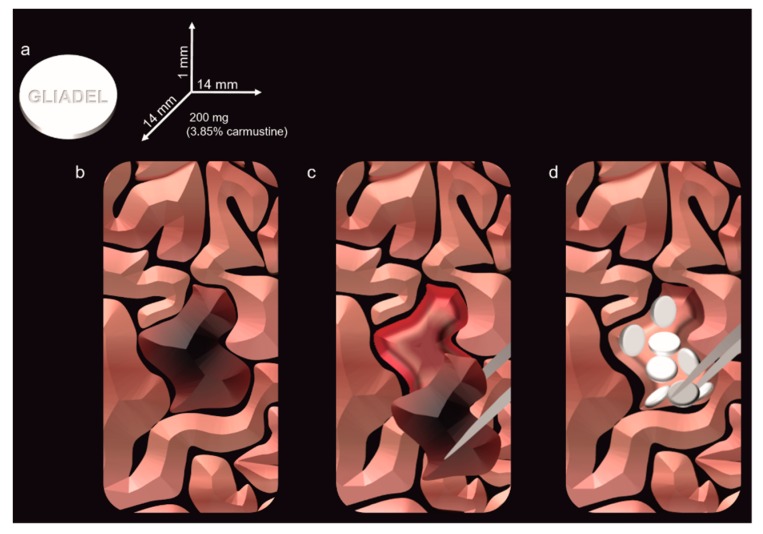
Gliadel wafers treatment: (**a**) physical and pharmaceutical features of Gliadel; (**b**) identification of the brain tumor, (**c**) maximal safe resection of the tumor tissue, (**d**) application of the Gliadel wafers in the tumor bed. Up to eight disks can be applied.

**Figure 5 pharmaceutics-11-00245-f005:**
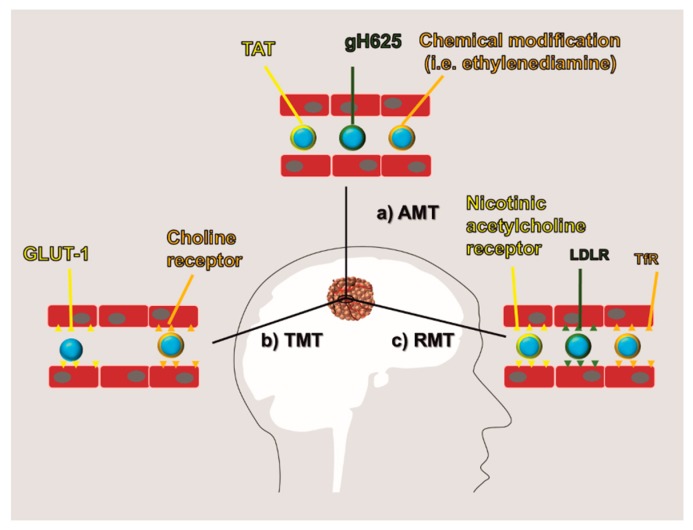
BBB negotiation: Current methods to negotiate BBB are obtained by modifying the therapeutic molecules or the carrier surface to increase their affinity for the BBB. They are generally referred to as: (**a**) adsorptive-mediated transcytosis (AMT) which is based on a positive surface charge of the therapeutics, (**b**) transporter-mediated transcytosis (TMT) which exploits the affinity of the therapeutics for endothelial transporters (i.e., GLUT1 and choline receptor), and (**c**) receptor-mediated transcytosis (RMT) which exploits the affinity of the therapeutics for endothelial receptors (i.e., nicotinic acetylcholine receptor, low-density lipoprotein receptor (LDLR), and transferrin receptor (TfR)).

**Figure 6 pharmaceutics-11-00245-f006:**
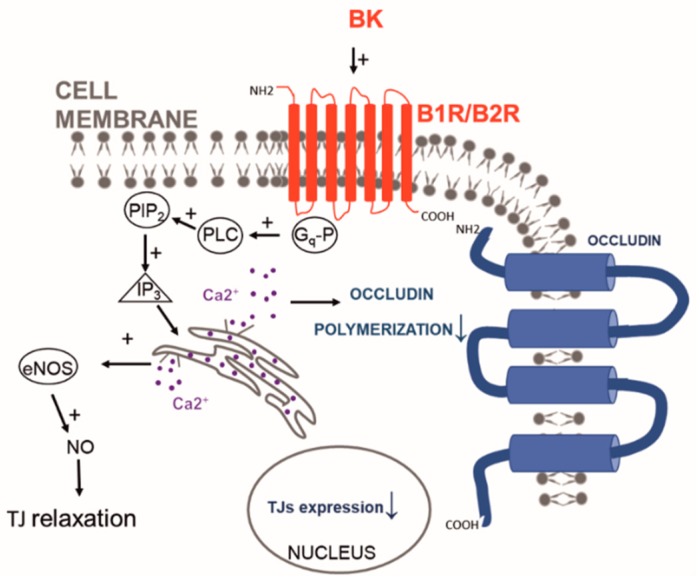
Bradykinin effect on TJs: bradykinin (BK), interacting with its receptors B1R and B2R, induces the intracellular release of Ca^2+^ that can affect occludin polymerization as well as increase the production of nitric oxide that relaxes the junctions. The activation of this pathway is also associated with a decrease in TJ mRNA expression.

**Figure 7 pharmaceutics-11-00245-f007:**
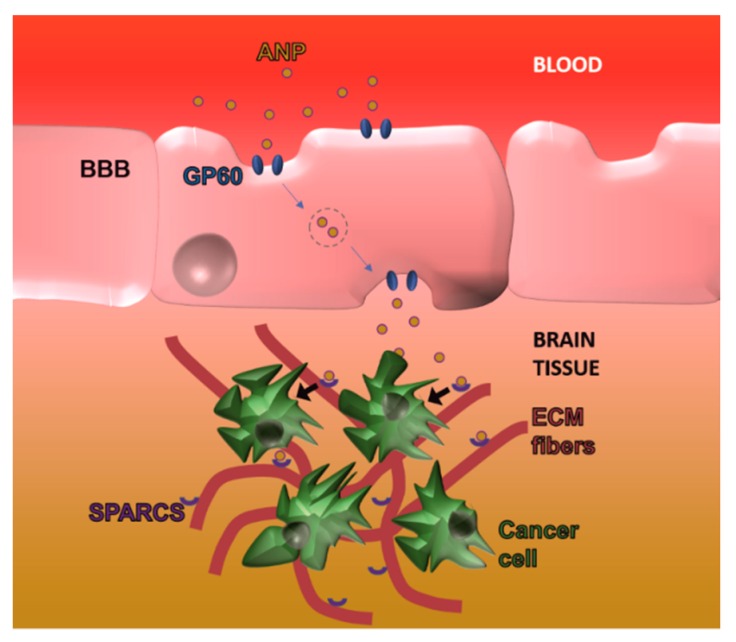
Exploiting tumor metabolic changes to overcome the BBB: Brain cancer lesions overexpress GP60 and secreted protein acidic and rich in cysteine (SPARC) that favor albumin nanoparticle trafficking over the BBB as well as cancer cell internalization in the abluminal side, respectively.

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
