# Peer review of "Established and Emerging Strategies for Drug Delivery Across the Blood-Brain Barrier in Brain Cancer"

_pharmaceutics, 2019, doi:10.3390/pharmaceutics11050245_

Round 1

Reviewer 1 Report

This review article is for the most part quite easy to read and informative. Some English language editing is required though. Some sections were very well written and comprehensive in their discussion, especially the early sections, and the FUS section. That then made other sections, where there was a lack of discussion of results from clinical or preclinical trials, disappointing. Additionally, while the title of the paper suggests discussion of drug delivery, there seems to be a focus on nanoparticles packaging up the drugs, rather than drug delivery per se. This added to the disappointment in sections such as 5.2, where no examples were discussed and the research was almost dismissed. Reading through the different sections, there isn’t a balance of research presented and almost a bias of work presented in some sections. Unfortunately, there needs to be some changes to remove this imbalance in some sections to reflect the research outputs in all areas of research regarding drug delivery to the brain.

Author Response

-This review article is for the most part quite easy to read and informative. Some English language editing is required though. Some sections were very well written and comprehensive in their discussion, especially the early sections, and the FUS section. That then made other sections, where there was a lack of discussion of results from clinical or preclinical trials, disappointing.

We thank the reviewer for her/his comments. We revised the entire document and adjusted many sections in their content and in the quality of the language.

-Additionally, while the title of the paper suggests discussion of drug delivery, there seems to be a focus on nanoparticles packaging up the drugs, rather than drug delivery per se.

We agree with the reviewer that a lot of the literature cited focused on nanomedicine and that drug delivery includes a larger field of discussion. However, in our review we also cited works that are not based on nanoparticles and more examples were included in the paper. Nanomedicine polarized the entire field of drug delivery over the last 2 decades, therefore we were almost obliged to include so many examples of research based on nanoparticles. 

-This added to the disappointment in sections such as 5.2, where no examples were discussed and the research was almost dismissed. Reading through the different sections, there isn’t a balance of research presented and almost a bias of work presented in some sections. Unfortunately, there need to be some changes to remove this imbalance in some sections to reflect the research outputs in all areas of research regarding drug delivery to the brain.

We did not realize the lack of balance between the different sections, and we thank the reviewer for this comment because it was fundamental to improve the paper. Section 5.2, as well as the introduction of section 3 and in general the overall document, were included with more recent research works to make the different sections more balanced between each other.

Reviewer 2 Report

This review article provides a comprehensive overview of advances in the field of drug delivery strategies across the blood-brain barrier, with an emphasis on brain cancer indications, which represents a unique challenge due to the presence of the blood-brain tumor barrier.

While sections that describe the various established and emerging approaches to overcome the BBB provide expert insights, sections that deal with the basics of the BBB demonstrate major shortcomings which are explained in my specific comments below.

l. 69. The statement “the BBB is defined as the complex net of blood capillaries located in the CNS” needs revision since also at other levels of the CNS vascular tree a functional BBB exists. Further, estimated total length and surface area in the following sentence requires citation of the appropriate reference by B. Zlokovic.

l. 79. The statement “the ability of the BBB to sort and eventually hamper the transport in the nervous tissue is due to specific tight junctions” is incorrect. BBB endothelial cells have a highly selective endo-lysosomal sorting mechanism that, depending on the substrate, either facilitates or complicates the transcellular transport across the endothelial layer. It’s not just the paracellular barrier that regulates transit. Please revise.

l. 81/82: Occludins come in 3 subtypes? Please provide a reference for this information. On a similar note, in line 85, the authors cite a paper referring to the role of “occludin-5” while they clearly mean “claudin-5”.

l. 91: Referring to TEER values greater than 1000 Ohms*cm2 as being a sufficient threshold for in vitro studies is simplified, and by the way deviates from the authors’ statement in line 157. Moreover, TEER does not only take tight junction-mediated (paracellular) flux of ions into account, but also the flux across the cell membrane (transcellular). A more refined definition as well as updated estimated values for physiological TEER is provided in DeStefano et al. Fluids Barriers CNS. 2018 Dec 4:15(1);32 and should be considered in this section for revision. In addition, the unit of TEER needs to be corrected (Ohms*cm2, not Ohms/cm2).

These comments also apply to the section on BBB models (l. 158)

l. 103+: Astrocytes not only associate with endothelial cells in the cerebellum, but throughout the brain. Moreover, astrocytes do not induce tight junction expression since they don’t appear at the BBB before birth; they are critical for BBB maintenance rather. It’s the pericytes that associate with developing nascent vessels and initiate tight junction expression, regulate transporter expression, and downregulate transcytosis as well as leukocyte adhesion molecules. Pericytes are therefore much more than “stabilizers” of BBB vessels. The authors are encouraged to revise this section based on the current knowledge from more recent research and review articles.

l. 157-159: The statement that TEER values of 150-200 need to be exceeded to qualify as in vitro BBB model needs revision. This is not in alignment with the current understanding in the field. See comments above. Further, an appropriate in vitro BBB model needs to demonstrate both tight junctions and transporter, not “and/or”.

l.188: The section on 3D models lacks a summary of state-of-the-art microfluidic platforms.

l. 240: Could the authors speculate on any potential long-term adverse effects on BBB endothelial cells when focused ultrasound is applied frequently, also taking into account the unwanted passage of blood-borne components into the neural tissue?

l. 410: AMT is adsorptive, not absorptive mediated transcytosis.

l. 459/460: None of the transporters and receptors currently being explored for drug delivery are “specifically expressed” at the BBB; they may be enriched at the BBB as compared to other peripheral vascular beds (such as GLUT-1, Transferrin Receptor). The wording should be changed to reflect this more accurately.

l. 463-465: A reference is needed for the concept that internalized cargo upon TMT/RMT interacts with receptors inside the cell, mediating the transcytosis to the abluminal side. I am not familiar with this mechanism.

l. 471: The section on RMT should be amended with examples of bispecific antibody engineering, usually referred to as Trojan Horse Technology for the delivery of biologics across the BBB.

l. 482: A reference is needed for the statement that modulating adherens junctions at the BBB “is not toxic”.

l. 562: Suggest changing title of this section to “Cell and Gene Therapy”.

Fig. 1: The legend is incorrect. Only pericytes (please correct typo) are embedded in the basement membrane, but not endothelial cells. The concept of “neuronal end-feet” requires a reference in which this term was used. Further, since the figure claims to represent the neurovascular unit, and to be consistent with the body text, microglia should be added.

Incorrect use of words or typos (this is not a complete list):

l. 32: epidimial; l. 101/102: membrane basement; l 138: perycites; l. 445: RTM ; l. 470: TFr ; l. 498+: LAPR; l. 519/520: syntax

Author Response

-This review article provides a comprehensive overview of advances in the field of drug delivery strategies across the blood-brain barrier, with an emphasis on brain cancer indications, which represents a unique challenge due to the presence of the blood-brain tumor barrier.

While sections that describe the various established and emerging approaches to overcome the BBB provide expert insights, sections that deal with the basics of the BBB demonstrate major shortcomings which are explained in my specific comments below.

We thank the reviewer for her/his extensive revision. We did our best to improve the first part of this paper and to edit all the mistakes and imprecisions we reported accordingly to the reviewer suggestions.

-I. 69. The statement “the BBB is defined as the complex net of blood capillaries located in the CNS” needs revision since also at other levels of the CNS vascular tree a functional BBB exists. Further, estimated total length and surface area in the following sentence requires citation of the appropriate reference by B. Zlokovic

We agree with the reviewer admitting that the definition of the BBB solely as a function of its location does not properly describe the importance and the specialization of this tissue. For this reason, we reported the definition provided by the group Dr. Zlokovic (Sweeeney et al.) and we cited their work in the statement regarding length and surface area of BBB (Current references 11 and 12).

-I. 79. The statement “the ability of the BBB to sort and eventually hamper the transport in the nervous tissue is due to specific tight junctions” is incorrect. BBB endothelial cells have a highly selective endo-lysosomal sorting mechanism that, depending on the substrate, either facilitates or complicates the transcellular transport across the endothelial layer. It’s not just the paracellular barrier that regulates transit. Please revise.

In this specific passage highlighted by the reviewer we may have superficially described the biological cues that determine the BBB properties. For this reason, we changes our statement on the biological causes determining BBB permeability reporting the importance of the intracellular sorting as suggested by the reviewer. However, the same concepts were extensively stated later in the text.

-I. 81/82: Occludins come in 3 subtypes? Please provide a reference for this information. On a similar note, in line 85, the authors cite a paper referring to the role of “occludin-5” while they clearly mean “claudin-5”.

We thank the reviewer for highlighting these mistakes and both the sentences were corrected. Occludin comes in 1 form and occludin-5 was changed in claudin-5. 

-Referring to TEER values greater than 1000 Ohms*cm2 as being a sufficient threshold for in vitro studies is simplified, and by the way deviates from the authors’ statement in line 157. Moreover, TEER does not only take tight junction-mediated (paracellular) flux of ions into account, but also the flux across the cell membrane (transcellular). A more refined definition as well as updated estimated values for physiological TEER is provided in DeStefano et al. Fluids Barriers CNS. 2018 Dec 4:15(1);32 and should be considered in this section for revision. In addition, the unit of TEER needs to be corrected (Ohms*cm2, not Ohms/cm2).  These comments also apply to the section on BBB models (I. 158)

More accurate information based on the work of DeStefano et al. were included. We referred as 900 Wxcm2 as reference for appropriate TEER since this value represents a cutoff for the IgG permeability of in vitro BBB models. We also provided a better definition of the ionic resistance registered by TEER according to the suggestion reviewer, and we corrected the unit of TEER.

-Astrocytes not only associate with endothelial cells in the cerebellum, but throughout the brain. Moreover, astrocytes do not induce tight junction expression since they don’t appear at the BBB before birth; they are critical for BBB maintenance rather. It’s the pericytes that associate with developing nascent vessels and initiate tight junction expression, regulate transporter expression, and downregulate transcytosis as well as leukocyte adhesion molecules. Pericytes are therefore much more than “stabilizers” of BBB vessels. The authors are encouraged to revise this section based on the current knowledge from more recent research and review articles.

We did change parts that needed revision as suggested. The comments of the reviewer are correct and also reported in the paper. However, astrocyte role in the expression of TJs is still under debate and while some papers claim they have a minor role in this process other suggest that astrocytes can definitely induce TJ expression both in vitro and in vivo, in particular in the adults. These concepts were reported in the new version of the review.

-157-159: The statement that TEER values of 150-200 need to be exceeded to qualify as in vitro BBB model needs revision. This is not in alignment with the current understanding in the field. See comments above. Further, an appropriate in vitro BBB model needs to demonstrate both tight junctions and transporter, not “and/or”.

This statement was changed referring simply to the need for registering high TEER values. A more specific explanation of the TEER was previously indicated in the text as the reviewer suggested while stating that both tight junction and transporter expression is necessary to validate in vitro BBB.

-The section on 3D models lacks a summary of state-of-the-art microfluidic platforms.

More information about 3D models were included.

-I. 240: Could the authors speculate on any potential long-term adverse effects on BBB endothelial cells when focused ultrasound is applied frequently, also taking into account the unwanted passage of blood-borne components into the neural tissue?

This is not the focus of this review and this kind of speculation can be applied also to other mechanisms of BBB breaching. We cited one of many works describing the toxicity of albumin on neurons and we implemented the FUS section with specific works evaluating the side effects of repetitive FUS. 

-I. 410: AMT is adsorptive, not absorptive mediated transcytosis.

The definition of AMT was amended accordingly to the reviewer suggestions. However, both of the terms are used in literature to describe this phenomenon. Here are some valuable citations:

-Doxorubicin-loaded nanoparticles consisted of cationic-and mannose-modified-albumins for dual-targeting in brain tumors HJ Byeon, S Lee, SY Min, ES Lee, BS Shin… - Journal of controlled release 2016

-Transferrin receptors-targeting nanocarriers for efficient targeted delivery and transcytosis of drugs into the brain tumors: a review of recent advancements and … H Choudhury, M Pandey, PX Chin, YL Phang… - Drug delivery and translational research, 2018 – Springer

- Targeted bioadhesive nanomedicine: an effective approach for synergistic drug delivery to cancers. MK Viswanadh, MS Muthu - 2018 - Future Medicine

-I. 459/460: None of the transporters and receptors currently being explored for drug delivery are “specifically expressed” at the BBB; they may be enriched at the BBB as compared to other peripheral vascular beds (such as GLUT-1, Transferrin Receptor). The wording should be changed to reflect this more accurately.

TfR was actually reported to be expressed only on BBB and not in normal endothelium and we reported in the text the proper citation as well as including more examples of research performed in the field. However, we deleted in this introductory sentence the words “specifically expressed”, since as highlighted by the reviewer the expression of transportes is generally enhanced in the BBB compared to other kinds of endothelium.

-I. 463-465: A reference is needed for the concept that internalized cargo upon TMT/RMT interacts with receptors inside the cell, mediating the transcytosis to the abluminal side. I am not familiar with this mechanism.

This concept was not properly expressed and needed for correction. Basically, we wanted to state that further luminal side targeting and release in the abluminal side, the therapeutic still needs to target cancer cells to be effective,  as it occurs for example for TfR that is overexpressed both on BBB and on cancer cells.

-I. 471: The section on RMT should be amended with examples of bispecific antibody engineering, usually referred to as Trojan Horse Technology for the delivery of biologics across the BBB.

We included this topic in the review

-I. 482: A reference is needed for the statement that modulating adherens junctions at the BBB “is not toxic”.

The advantage of targeting Adherens junctions from the standpoint of the toxicity is that disrupting the continuity of these proteins is followed by a relatively fast and controlled reversibility. This concept and corresponding reference was used to replace the sentence highlighted by the reviewer.

-I. 562: Suggest changing title of this section to “Cell and Gene Therapy”.

We changed the title of Section 7

-Fig. 1: The legend is incorrect. Only pericytes (please correct typo) are embedded in the basement membrane, but not endothelial cells. The concept of “neuronal end-feet” requires a reference in which this term was used. Further, since the figure claims to represent the neurovascular unit, and to be consistent with the body text, microglia should be added.

The legend was corrected and microglia cells added. The end-feet related to the neuron was a mistake and deleted and the legend was amended.

-Incorrect use of words or typos (this is not a complete list):

I. 32: epidimial; l. 101/102: membrane basement; I 138: perycites; I. 445: RTM ; I. 470: TFr ; I. 498+: LAPR; I. 519/520: syntax

We corrected these typos.

Round 2

Reviewer 1 Report

The authors have amended the article and I am happy to accept in its current form.